# Semi-LAR: Semi-supervised Contrastive Learning with Linear Attention for Removal of Nighttime Flares

**Xiyu Zhu**[1] **Wei Wang**[*1] **Kui Jiang**[2] **Zhengguo Li**[3]

## Abstract

Lens flare removal is challenging due to the large spatial extent of flare artifacts and their entanglement with scene structures, while existing methods heavily rely on large-scale paired data. We propose a semi-supervised flare removal framework that enables stable learning from unlabeled images by jointly addressing pseudo-label reliability and representation discrimination. We propose an adaptive pseudo-label repository that progressively refines pseudo supervision through no-reference quality assessment, momentum-based updates, and invalid label filtering, effectively mitigating error accumulation. Moreover, we propose a flare-aware contrastive loss that explicitly treats flare-contaminated inputs as negatives and performs patch-level contrastive learning, encouraging representations that are discriminative against flare patterns while remaining consistent with reliable pseudo targets. Extensive experiments on multiple flare benchmarks demonstrate that the proposed framework is model-agnostic and consistently improves performance and robustness.

## 1. Introduction

Lens flare is a common artifact which is produced when strong light sources interact with a camera's optical system (Ernst et al., 2005; Hullin et al., 2011). In night scenes, flare often appears as large-area halos, radial streaks, veiling glare, and ghost reflections. These effects reduce contrast and introduce color shifts, which not only harms visual qual-

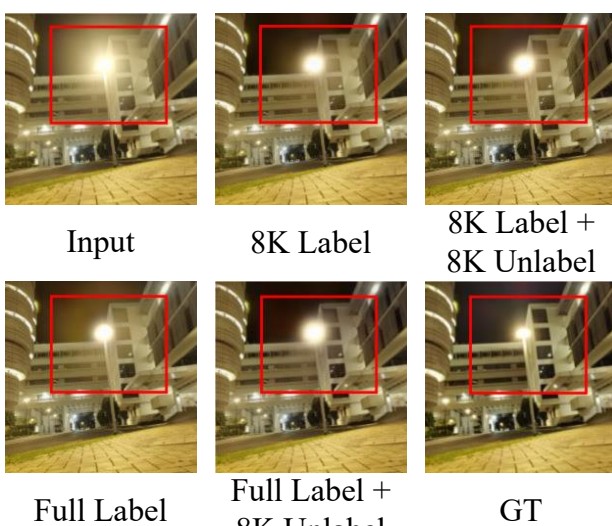

Input — 8K Label — 8K Label + 8K Unlabel

Full Label — Full Label + 8K Unlabel — GT

*Figure 1.* Our approach bridges the gap to full supervision under limited labels and achieves further quality improvements when added to existing labeled baselines.

ity but can also disrupt downstream vision tasks that rely on stable edges, consistent illumination, and accurate color statistics (Li et al., 2025b; Girshick et al., 2014). Similar nighttime visibility degradation has also been studied from the perspective of irregular glow removal and glow-aware enhancement (Wu et al., 2024b).Because flare patterns vary with lens design, aperture and exposure, removing flare while keeping the light source and scene details intact remains a challenging problem (Qiao et al., 2021). In practical nighttime imaging, lens flare may also coexist with other degradations such as camera shake and spatio-temporally varying motion blur in videos (Li et al., 2024), making robust visual restoration more challenging.

Earlier approaches primarily relied on hardware-based solutions (Raut et al., 2011) or classical image processing techniques. Typical pipelines detected flare regions using hand-crafted cues such as intensity thresholds, saturation patterns, or edge responses, followed by filtering, decomposition, or inpainting to restore corrupted areas (Vitoria & Ballester, 2019; Xie & Tu, 2015). Although these methods can handle simple glare patterns, they are often brittle in

*Equal contribution [1]School of Computer Science and Technology, Wuhan University of Science and Technology, Wuhan, China [2]Faculty of Computing, Harbin Institute of Technology, Harbin 150001, China [3]SRO department, Institute for Infocomm Research, A*STAR, Singapore. Correspondence to: Wei Wang <wangwei8@wust.edu.cn>.

*Proceedings of the 43rd International Conference on Machine Learning*, Seoul, South Korea. PMLR 306, 2026. Copyright 2026 by the author(s).

practice: flare frequently overlaps with legitimate highlights or reflections, and complex flare structures are difficult to model with fixed rules or parametric assumptions.

With the rise of deep learning, flare removal has shifted to supervised learning with paired data, and progress has been closely linked to datasets and synthesis strategies. Early work enabled large-scale training through physically motivated synthesis (Wu et al., 2021b), followed by benchmarks such as Flare7K and Flare7K++ (Dai et al., 2022; 2023a), which improved flare diversity and realism. More recent datasets, including FlareX (Qu et al., 2025) and PBFG (Zhu & Lee, 2025), further emphasize physical plausibility and challenging flare distributions. Meanwhile, model architectures have evolved toward transformer-based and frequency-aware designs to capture large-area structures and long-range dependencies. Despite these advances, most methods still rely heavily on large paired datasets, which are costly and limit scalability.

To leverage large-scale real-world flare-corrupted images without ground truth, semi-supervised learning has emerged as a promising solution (He et al., 2025). However, flare removal faces two key challenges: noisy pseudo targets that destabilize training, and confusion between flare patterns and true highlights, which may lead to over-smoothing or removal of legitimate structures (Jin et al., 2023b).

To address these challenges, we propose a semi-supervised flare removal framework built on reliable pseudo supervision and flare-aware representation learning. The framework introduces a patch-level contrastive loss that explicitly separates restored features from flare-contaminated inputs while maintaining consistency with pseudo targets. In addition, an adaptive pseudo-label $y$ is designed to progressively refine pseudo supervision using MUSIQ-based (Ke et al., 2021) quality assessment and momentum updates. The proposed approach is model-agnostic and can be seamlessly integrated into existing restoration backbones, consistently improving performance and robustness.

The main contributions of this paper are summarized as follows:

- We propose a flare-aware patch-level contrastive learning strategy that explicitly separates flare-contaminated inputs from pseudo targets in the representation space, leading to more stable training and fewer flare-related artifacts.

- We design an adaptive pseudo-label repository guided by no-reference quality assessment and momentum updates, which effectively reduces error accumulation in semi-supervised flare removal.

- We present a general semi-supervised framework that can be directly applied to mainstream image restora-

tion models and demonstrate consistent performance improvements over strong fully supervised baselines on multiple flare benchmarks.

## 2. Related Work

**Transformer-based Architectures for Flare Removal.** Recent flare removal methods have evolved to better capture the global and non-local nature of flare artifacts. Transformer-based frameworks are widely adopted due to their ability to model long-range dependencies and large-area degradations. Sparse-UFormer (Wu et al., 2024a) introduces sparsity-aware attention to focus computation on flare-relevant regions, while LPFSFormer (Chen et al., 2024) incorporates explicit location priors to guide frequency–spatial interaction. Several works further enhance transformer backbones with task-specific cues. Flare-Free Vision injects depth information into a Uformer-based architecture to disambiguate flare artifacts from scene content (Kotp & Torki, 2024), while LUFormer (Lu et al., 2025) leverages luminance-aware localized attention and frequency augmentation to better handle nighttime flare intensity variations. These approaches demonstrate that integrating auxiliary priors and domain-specific knowledge into transformer architectures is crucial for improving flare suppression performance in challenging real-world scenarios.

**Frequency domain Model of Flare Artifacts.** Frequency domain modeling has also become a prominent design choice. FF-Former (Zhang et al., 2023), DFDNet (Xue et al., 2025), and SLCFormer (Zhu et al., 2025) explicitly integrate Fourier-domain processing to capture global illumination shifts and structured flare patterns. MFDNet (Jiang et al., 2024) decomposes images into multiple frequency bands to achieve efficient flare suppression with reduced computational cost, while FCNet (Qi et al., 2025a) employs feature-complementary branches to balance local detail preservation and global flare removal. More recent works explore hybrid spectral–spatial interaction and adaptive frequency guidance to improve robustness under uneven flare distortion (Liu et al., 2025).

**Incorporating Additional Priors.** Beyond conventional image-to-image restoration, several studies incorporate additional priors or alternative representations. SGSFT (Ma et al., 2025) guided approaches exploit internal image statistics to extract flare-related priors without external annotations, while SPDDNet (Qi et al., 2025b) based dual-domain networks introduce prompt-like guidance to steer restoration in both spatial and frequency domains. Flare-Aware RWKV (Zhang et al., 2025) further explores sequence modeling paradigms to capture long-range flare dependencies with reduced attention overhead. In addition to flare-specific restoration, low-light image enhancement has also

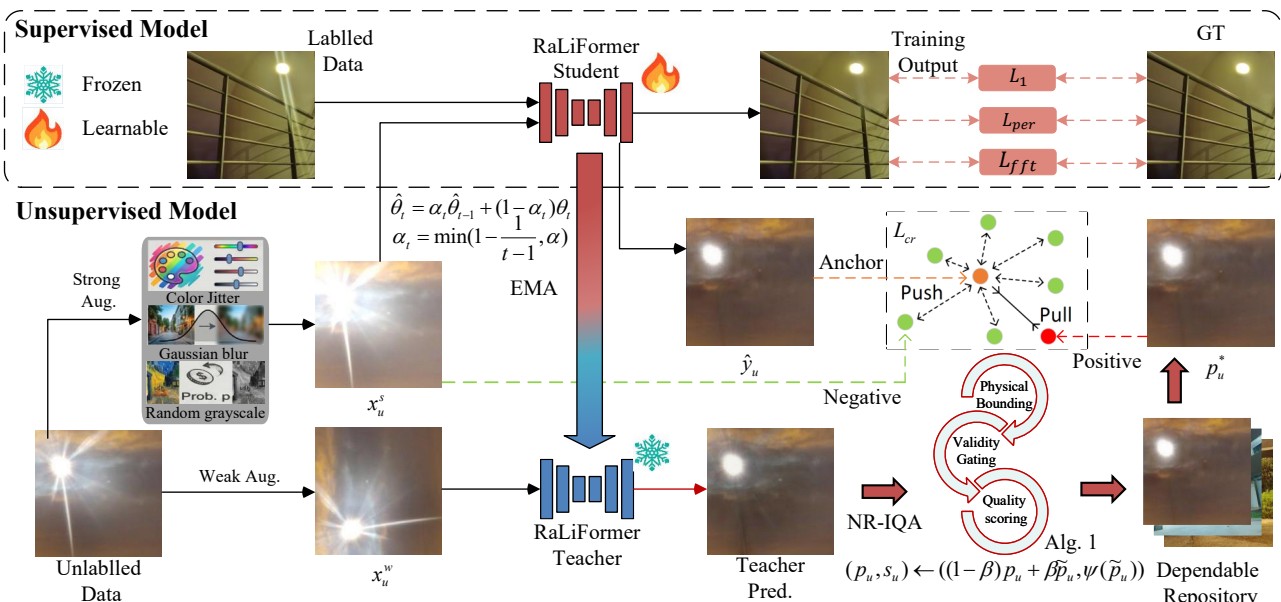

*Figure 2.* Overview of the proposed semi-supervised flare removal framework. The method adopts a teacher–student paradigm with EMA-based teacher updating. Labeled data are optimized with supervised reconstruction losses, while unlabeled data are processed using weak and strong augmentations. Teacher predictions are evaluated and selectively stored in a dependable repository to provide reliable pseudo supervision. The student is trained with a consistency objective augmented by flare-aware contrastive regularization.

been explored with lightweight CNN designs and FPGA-based deployment for efficient nighttime visibility improvement (Wang & Xu, 2023). However, these methods mainly focus on illumination enhancement and hardware efficiency, rather than structured lens flare removal. Despite these advances, most existing approaches remain fully supervised and rely heavily on large-scale paired datasets, limiting scalability and robustness when confronted with diverse real-world flare distributions.

**Semi-supervised learning for vision tasks.** Semi-supervised learning aims to leverage abundant unlabeled data to improve generalization when labeled samples are scarce. Early paradigms such as pseudo-labeling and self-training treat high-confidence predictions on unlabeled data as additional supervision, enabling iterative refinement of model parameters (Lee et al., 2013). Consistency-based approaches further encourage prediction invariance under input perturbations, forming the foundation of many modern methods. In large-scale recognition, Noisy Student (Xie et al., 2020) demonstrates that injecting noise through data augmentation, stochastic depth, and dropout during self-training can substantially improve robustness and accuracy. Beyond high-level tasks, semi-supervised learning has also been explored in low-level vision problems, where paired supervision is difficult to obtain. Methods like (Jin et al., 2023a) and (Cong et al., 2024) show that incorporating domain-specific priors and frequency-aware constraints can effectively stabilize semi-supervised training and improve

perceptual quality under challenging illumination conditions.

**Teacher–student and consistency-based frameworks.**
Teacher–student learning is a widely used paradigm in semi-supervised learning, with Mean Teacher (Tarvainen & Valpola, 2017) serving as a representative framework. By updating the teacher as an exponential moving average of the student, it enforces prediction consistency on unlabeled data and provides stable supervision without modifying network architectures. In low-level vision tasks, where outputs are continuous and pseudo-label noise is particularly harmful, recent studies combine consistency learning with task-specific regularization. WDUIE (Qian et al., 2025) stabilizes semi-supervised enhancement via wavelet-domain diffusion modeling, while Semi-LLIE (Li et al., 2025a) introduces contrastive learning to improve representation discrimination in low-light enhancement. However, despite the abundance of unlabeled real-world flare images, semi-supervised learning has been rarely explored in lens flare removal. This gap suggests significant potential for leveraging semi-supervised and weakly supervised strategies tailored to the unique characteristics of flare artifacts.

## 3. Method

In this section, we introduce the semi-supervised flare removal framework with reliable pseudo supervision and flare-aware representation learning (Figure 2). The framework

consists of a dependable pseudo-label repository and and a flare-aware contrastive regularization, together with the generator architecture.

### 3.1. Preliminary

For labeled training pairs, although flare-free references can be obtained from real captures, we follow the same synthesis pipeline as Flare7K++ (Dai et al., 2023a) to construct paired supervision at scale. Specifically, a clean background image is combined with a randomly transformed flare pattern, together with photometric perturbations.

We adopt the teacher-student learning paradigm (Tarvainen & Valpola, 2017) implemented by two separate network instances with identical architectures. The student network is optimized by gradient descent, while the teacher network serves as a stable target generator and is updated only through exponential moving average (EMA) of the student parameters:

$$\bar{\theta} \leftarrow \alpha \bar{\theta} + (1 - \alpha) \theta, \tag{1}$$

where $\theta$ and $\bar{\theta}$ denote the parameters of the student and teacher networks, respectively, and $\alpha$ controls the update smoothness.

Under this formulation, the student is supervised by ground-truth targets on labeled data and by teacher-generated pseudo supervision on unlabeled data. The overall training objective can be summarized as

$$\mathcal{L}_{\text{total}} = \mathcal{L}_{\text{sup}} + \eta \, \mathcal{L}_{\text{unsup}}, \tag{2}$$

where $\mathcal{L}_{\text{sup}}$ denotes the supervised loss on paired samples, $\mathcal{L}_{\text{unsup}}$ enforces teacher-student consistency on unlabeled data, and $\eta$ balances the two terms. The detailed definitions of these losses and the mechanisms for reliable pseudo supervision are introduced in the following sections. For unlabeled data, we generate two views via data augmentation: a weakly augmented view $x_u^w$ for teacher prediction and a strongly augmented view $x_u^s$ for student optimization.

### 3.2. NR-IQA Based Dependable Repository for Reliable Consistency

To exploit unlabeled flare images without accumulating erroneous pseudo supervision, we maintain a *dependable repository* that stores a persistent pseudo target for each unlabeled sample $x_u$ and updates it only when the newly generated candidate is *reliably better*. For each unlabeled image, the repository keeps a tuple $(p_u, s_u)$, where $p_u$ denotes the stored pseudo label and $s_u$ is its NR-IQA score estimated by MUSIQ $\Psi(\cdot)$ (Ke et al., 2021). At each iteration, the teacher produces a candidate prediction $\hat{p}_u$ on the $x_u^w$, which is first bounded by a lightweight physical plausibility constraint and then evaluated by MUSIQ. The repository is updated only if the candidate (i) passes validity

---

**Algorithm 1** Update of Dependable Repository (DR)
___
**Require:** NR-IQA $\Psi(\cdot)$; batch of unlabeled weak views $\{x_{u,i}^w\}_{i=1}^B$; teacher predictions $\{\hat{p}_{u,i}\}_{i=1}^B$; stored pseudo labels $\{p_{u,i}\}$ and scores $\{s_{u,i}\}$
1: **for each** $i = 1, \ldots, B$ **do**
2:     $\tilde{p}_{u,i} \leftarrow \text{clip}(\min(\hat{p}_{u,i}, x_{u,i}^w + \epsilon), 0, 1)$
3:     $is\_invalid \leftarrow (\mu(\tilde{p}_{u,i}) < \tau_{\text{black}}) \vee (\text{amin}(\tilde{p}_{u,i}) > \tau_{\text{fog}})$
4: **if** $is\_invalid$ **then**
5:         **continue**
6: **end if**
7:     $s_t \leftarrow \Psi(\tilde{p}_{u,i})$
8:     $is\_empty \leftarrow (\mu(p_{u,i}) < \tau_{\text{empty}})$
9:     $is\_better \leftarrow (s_t > s_{u,i} + \delta)$
10: **if** $is\_better \vee is\_empty$ **then**
11:         $p_{u,i} \leftarrow (1 - \beta)p_{u,i} + \beta\tilde{p}_{u,i}$
12:         $s_{u,i} \leftarrow s_t;$
13: **end if**
14: **end for**
    Updated DR $\{p_{u,i}\}$ and $\{s_{u,i}\}$
___

gating and (ii) improves the stored quality score by a margin (or the stored target is invalid/uninitialized). This update rule can be compactly summarized as The overall update rule of the dependable repository can be summarized as:

$$(p_u, s_u) \leftarrow \big((1-\beta)p_u + \beta\tilde{p}_u, \ \Psi(\tilde{p}_u)\big), \quad \text{if } \Psi(\tilde{p}_u) > s_u + \delta. \tag{3}$$

where $\tilde{p}_u$ denotes the physically bounded and validity-checked candidate, and $\beta$ controls the update smoothness of the stored pseudo-label. and the full procedure is given in Algorithm 1.

During training, a separate validity mask is applied when computing the unsupervised loss to exclude near-black pseudo-labels retrieved from the repository. This additional filtering step improves optimization stability and complements the repository-level gating by preventing rare degenerate cases from contributing gradients.

### 3.3. Flare-Aware Contrastive Regularization

Consistency-based pseudo supervision is insufficient for flare removal, as it may inadvertently preserve residual flare patterns and amplify teacher errors. To explicitly suppress confirmation bias, we propose a flare-aware contrastive regularization that treats flare-contaminated inputs as structured negatives and enforces discriminative representation learning.

**Patch-level contrastive formulation.** Let $f_\theta$ denote the student network and $\phi(\cdot)$ an intermediate feature extractor. For an unlabeled sample, the student predicts a restored

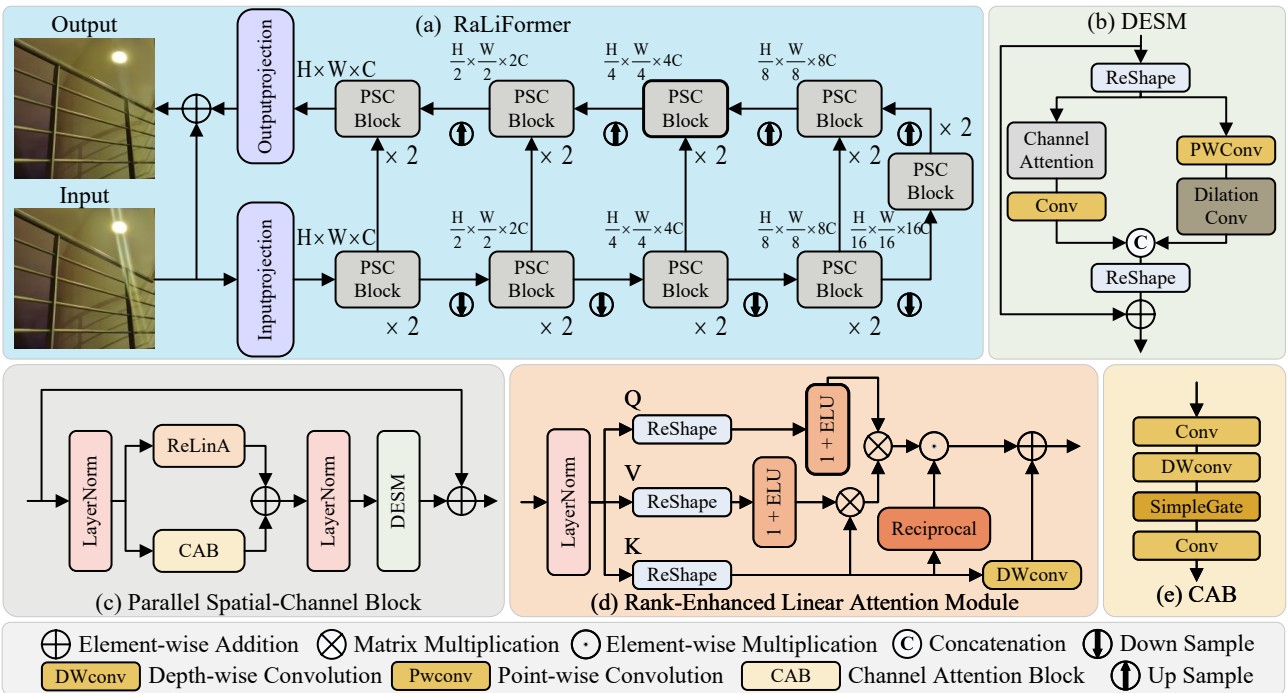

*Figure 3.* Overview of the proposed RaLiFormer architecture. (a) The hierarchical U-shaped network structure for image restoration. (b) The Directionally-Enhanced Spatial Module (DESM). (c) The Parallel Spatial-Channel Block (PSCBlock) as the basic building unit. (d) The Rank-Enhanced Linear Attention (ReLinA) module. (e) Detailed structure of the Channel Attention Block module.

output $\hat{y}_u = f_\theta(x_u^s)$, while a reliable pseudo $p_u^*$ is retrieved from the dependable repository. Following prior successes of contrastive learning in semi-supervised learning (Chen et al., 2020; Wu et al., 2021a), we incorporate the flare-aware contrastive formulation. We extract patch-level features from these images, yielding anchor features $\mathbf{z}_u^a = \phi(\hat{y}_u)$, positive features $\mathbf{z}_u^+ = \phi(p_u^*)$, and negative features $\mathbf{z}_u^- = \phi(x_u^s)$. Patch-wise construction allows abundant contrastive pairs even with small batch sizes and aligns well with the local nature of flare artifacts.

**Flare-aware contrastive loss.** For each anchor feature $\mathbf{z}_u^a$, we encourage similarity to the reliable pseudo target while discouraging similarity to the flare-corrupted input. The contrastive objective is defined as

$$\mathcal{L}_{\text{cr}} = -\log \frac{\exp\left(\text{sim}(\mathbf{z}_u^a, \mathbf{z}_u^+)/\tau\right)}{\sum_{v \in \{+,-\}} \exp\left(\text{sim}(\mathbf{z}_u^a, \mathbf{z}_u^v)/\tau\right)}, \quad (4)$$

where $\text{sim}(\cdot, \cdot)$ denotes cosine similarity and $\tau$ is a temperature parameter. In practice, one or more negative patches can be sampled from the flare-corrupted input.

**Closed-loop interaction with the dependable repository.** The proposed contrastive regularization is not used independently from the dependable repository. Instead, the two components form a closed-loop training mechanism. The repository provides progressively refined pseudo targets

to make the contrastive objective reliable, while the flare-aware contrastive loss encourages the student to separate flare-contaminated representations from underlying scene textures, which in turn improves the quality of subsequent teacher predictions and repository updates.

### 3.4. RaLiFormer Generator Structure

Our framework adopts a transformer-based flare removal model, named RaLiFormer, as the generator for both teacher and student networks. The proposed semi-supervised strategy is model-agnostic and introduces no architectural modification during inference. RaLiFormer follows an encoder–decoder design with multi-scale feature extraction and skip connections. More details are shown in Figure 3.

**Dual-attention building block.** The generator is built upon a dual-attention block that jointly captures global spatial dependencies and channel-wise feature adaptation. Given an intermediate feature map $X \in \mathbb{R}^{H \times W \times C}$, the block adopts a residual formulation:

$$X' = X + \mathcal{A}(X) + \mathcal{G}(X), \quad (5)$$

where $\mathcal{A}(\cdot)$ models long-range spatial interactions and $\mathcal{G}(\cdot)$ performs channel-adaptive feature refinement.

**Rank-Enhanced Linear Attention (ReLinA).** The spatial attention branch is implemented using Rank-Enhanced

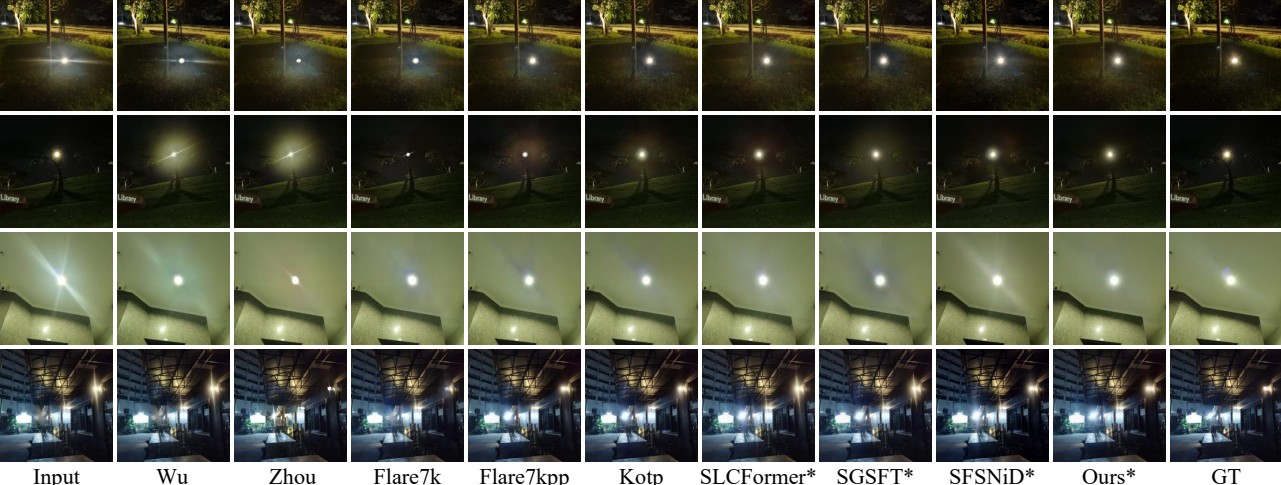

| Input | Wu | Zhou | Flare7k | Flare7kpp | Kotp | SLCFormer* | SGSFT* | SFSNiD* | Ours* | GT |

*Figure 4.* Visual comparison of flare removal on real-world nighttime flare images. * indicates models trained with 8K labeled and 8K unlabeled data.

Linear Attention (ReLinA), which enables efficient global context modeling with linear complexity. Given query, key, and value projections $(\mathbf{Q}, \mathbf{K}, \mathbf{V})$, a positive feature mapping is applied:

$$\mathbf{Q}' = \phi(\mathbf{Q}), \quad \mathbf{K}' = \phi(\mathbf{K}), \quad \phi(x) = 1 + \mathrm{ELU}(x). \quad (6)$$

The attention output is computed as:

$$\mathcal{A}(X) = \frac{\mathbf{Q}' \left( \mathbf{K}'^{\top} \mathbf{V} \right)}{\mathbf{Q}' \left( \sum_j \mathbf{K}'_j \right) + \varepsilon}, \quad (7)$$

where $\varepsilon$ ensures numerical stability. ReLinA is further coupled with convolutional refinement to enhance local structural representation.

**Directionally-Enhanced Feed-Forward Network.** Inspired by the DESM design in SLCFormer (Zhu et al., 2025), each block further incorporates the DESM as the feed-forward component, which introduces non-linear transformations and spatially-aware feature modulation via lightweight directional convolutions and gated interactions.

Due to space limitations, detailed architectural configurations and implementation specifics are provided in the supplementary material.

### 3.5. Overall Optimization Objective

The student network is trained by jointly optimizing supervised and unsupervised objectives within the teacher-student framework. The overall loss is defined as Eq. 2.

For labeled samples, supervision is applied to the restored output $\hat{y}_l$. We employ a weighted combination of pixel-wise, perceptual, and frequency-domain constraints:

$$\mathcal{L}_{\mathrm{sup}} = \lambda_1 \, \mathcal{L}_1(\hat{y}_l, y_l) + \lambda_2 \, \mathcal{L}_{\mathrm{per}}(\hat{y}_l, y_l) + \lambda_3 \, \mathcal{L}_{\mathrm{FFT}}(\hat{y}_l, y_l), \quad (8)$$

where the $\mathcal{L}_1$ term enforces pixel-level fidelity, the perceptual loss preserves structural consistency, and the FFT loss regularizes frequency discrepancies caused by flare artifacts. Here, $\hat{y}_l$ and $y_l$ denote the restored result and the corresponding ground-truth clean image of the labeled sample, respectively.

For unlabeled samples, reliable pseudo targets $p_u^*$ retrieved from the dependable repository are used as supervision. Given the student prediction $\hat{y}_u$ on the strong-view input, the unsupervised objective is formulated as

$$\mathcal{L}_{\mathrm{unsup}} = \lambda_1 \, \mathcal{L}_1(\hat{y}_u, p_u^*) + \lambda_{\mathrm{cr}} \, \mathcal{L}_{\mathrm{cr}}, \quad (9)$$

where $\mathcal{L}_{\mathrm{cr}}$ denotes the proposed flare-aware contrastive loss. The contrastive term plays a key role in suppressing confirmation bias by explicitly discouraging flare-related representations under pseudo supervision.

## 4. Experimental Results

### 4.1. Implementation Details

All experiments are implemented in PyTorch and conducted on a single NVIDIA RTX 3090 GPU. During training, images are randomly cropped to $512 \times 512$, with a batch size of 4. The student network is optimized using the Adam optimizer with momentum parameters $\beta_1 = 0.9$ and $\beta_2 = 0.99$, with an initial learning rate of $1 \times 10^{-4}$, while the teacher network is updated via exponential moving average. Due to space limitations, more implementation details and hyperparameter settings are provided in the supplementary material.

*Table 1.* Cross-dataset evaluation results trained on a partial Flare7K++ dataset and tested on the FlareX benchmark.

| Model | Dataset | PSNR | G-PSNR | LPIPS |
|---|---|---|---|---|
| Flare7K | Full | 19.496 | 29.765 | 0.140 |
| Flare7Kpp | Full | 19.007 | 28.749 | 0.143 |
| Flare-Free | Full | 19.232 | 29.038 | 0.140 |
| DeflareMamba | Full | 18.816 | 28.625 | **0.138** |
| **Ours** | 8K + 8K | **19.960** | **29.899** | 0.141 |

*Table 2.* Ablation of core components in our semi-supervised framework.

| Variant | PSNR | SSIM | LPIPS | G-PSNR | S-PSNR |
|---|---|---|---|---|---|
| w/o both | 26.541 | 0.879 | 0.0559 | 23.130 | 21.341 |
| w/o DR | 26.746 | 0.885 | 0.0533 | 23.427 | 21.658 |
| w/o $\mathcal{L}_{cr}$ | 26.711 | 0.884 | 0.0538 | 23.297 | 21.667 |
| Full model | **27.620** | **0.897** | **0.0493** | **24.276** | **22.686** |

### 4.2. Datasets and Metrics

**Datasets.** To evaluate performance under limited supervision, we construct the training set using only a subset of background images from Flickr-24K (Zhang et al., 2018b). Specifically, the first 8,000 images are used to synthesize labeled training pairs, while images from 8,000 to 16,000 are used solely as unlabeled data, with no overlap between the two subsets. Paired samples are synthesized following the Flare7K++ (Dai et al., 2023a) protocol, including geometric and photometric augmentations. Consistent with Flare7K++, we exclude 5,000 scattering flares from Flare7K and 964 samples from Flare-R, enabling fair comparison with existing methods while evaluating data efficiency under limited supervision.

**Evaluation Metrics.** To thoroughly evaluate the flare removal performance, we employ a set of complementary restoration metrics, including PSNR, SSIM (Wang et al., 2004), and the perceptual similarity metric LPIPS (Zhang et al., 2018a). In addition to these general-purpose measures, we adopt two region-aware metrics proposed by (Dai et al., 2023a), namely S-PSNR and G-PSNR, which are specifically designed to assess restoration quality in localized streak and glare regions, respectively.

### 4.3. Comparison with the State-of-the-Arts

**Compared methods.** For a comprehensive comparison, we evaluate our method against representative flare removal models under different supervision regimes, which can be grouped into three categories. (1) *Early learning-based methods*, including those proposed by (Wu et al., 2021b) and Zhou (Zhou et al., 2023). (2) *Fully supervised deep*

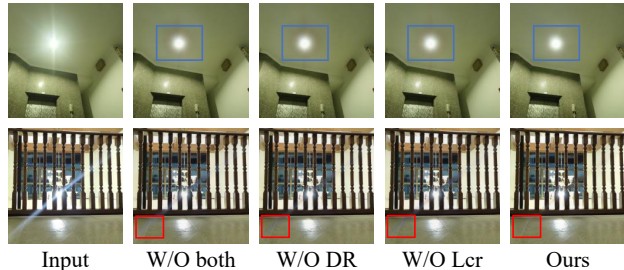

Input   W/O both   W/O DR   W/O Lcr   Ours

*Figure 5.* Ablation studies on the proposed method.

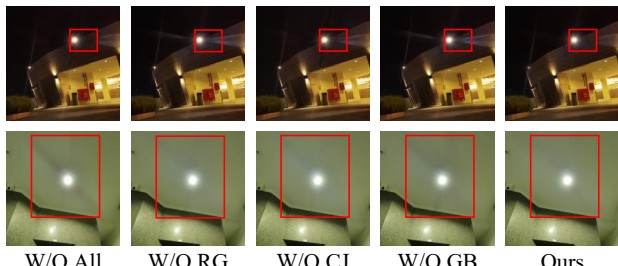

W/O All   W/O RG   W/O CJ   W/O GB   Ours

*Figure 6.* Ablation studies on the impact of strong augmentations .

*learning methods* trained on the full Flare7K++ dataset, such as U-Net (Ronneberger et al., 2015), Restormer (Zamir et al., 2022), Uformer (Wang et al., 2022), DeflareMamba (Huang et al., 2025), and SGSFT (Ma et al., 2025), which serve as strong upper-bound baselines. (3) *Limited-supervision and semi-supervised methods*, including limited-data variants of Uformer, Flare-Free Vision, SGSFT, and SLCFormer trained with 8K labeled samples, as well as recent semi-supervised approaches such as SFSNiD (Cong et al., 2024). Our method falls into this category and is trained with 8K labeled and 8K unlabeled images.

**Evaluation datasets.** All quantitative evaluations are conducted on the Flare7K++ (Dai et al., 2023a) and FlareX (Qu et al., 2025) test sets. In addition, representative qualitative results are reported on FlareReal600 (Dai et al., 2023b) for visual comparison.

**Objective comparison.** When the labeled data is reduced to 8K, the performance of fully supervised models drops noticeably, highlighting their reliance on large-scale paired data. In contrast, our semi-supervised framework effectively mitigates this degradation by exploiting additional unlabeled images. With only 8K labeled samples (about one third of the full supervision) and 8K unlabeled data, our method retains approximately **96%** of the fully supervised state-of-the-art performance in terms of PSNR and SSIM. Similar advantages are also observed on the FlareX test set (Table 1), further validating the improved data efficiency and robustness of the proposed approach. To further evaluate robustness beyond synthetic testing distributions, we

*Table 3.* Comparison with state-of-the-art flare removal methods. "Limited-Sup." denotes models trained using the same limited labeled set as ours. The highlighted full-supervision are shown to illustrate that our model trained with partial data attains comparable performance. Best and second-best results among limited- and semi-supervised methods are highlighted in bold and underline, respectively.

| Method | Supervision | Training Data | PSNR↑ | SSIM↑ | LPIPS↓ | G-PSNR↑ | S-PSNR↑ |
|---|---|---|---|---|---|---|---|
| Wu | Full | Previous Data | 24.613 | 0.871 | 0.0598 | 21.772 | 16.728 |
| Zhou | Full | Previous Data | 25.149 | 0.883 | 0.0576 | 22.053 | 17.865 |
| Flare7k | Full | Full Flare7K | 26.978 | 0.890 | 0.0466 | 23.828 | 21.563 |
| U-Net | Full | Full Flare7Kpp | 27.189 | 0.894 | 0.0452 | 23.527 | 22.647 |
| HINet | Full | Full Flare7Kpp | 27.548 | 0.892 | 0.0464 | 24.081 | 22.907 |
| Restormer | Full | Full Flare7Kpp | 27.597 | 0.897 | 0.0447 | 23.828 | 22.452 |
| Uformer | Full | Full Flare7Kpp | 27.633 | 0.894 | 0.0428 | 23.949 | 22.603 |
| Flare-Free | Full | Full Flare7Kpp | 27.662 | 0.897 | 0.0422 | 23.987 | 22.847 |
| DeflareMamba | Full | Full Flare7Kpp | 27.789 | 0.899 | 0.0449 | 24.406 | 22.378 |
| SGSFT | Full | Full Flare7Kpp | 28.077 | 0.904 | 0.0416 | 24.477 | 23.305 |
| SLCFormer | Full | Full Flare7Kpp | 28.092 | 0.905 | 0.0400 | 24.497 | 23.287 |
| Flare-Free | Limited-Sup. | 8K labeled | 27.240 | 0.886 | 0.0472 | 23.564 | 22.305 |
| SGSFT | Limited-Sup. | 8K labeled | 27.601 | 0.896 | 0.0497 | 24.201 | 22.665 |
| SLCFormer | Limited-Sup. | 8K labeled | 27.301 | 0.889 | 0.0477 | 23.920 | 22.052 |
| SFSNiD | Semi-Sup. | 8K labeled + 8K unlabeled | 24.992 | 0.860 | **0.0446** | 23.178 | 21.520 |
| **Ours** | Semi-Sup. | 8K labeled + 8K unlabeled | **27.620** | **0.897** | 0.0493 | **24.276** | **22.686** |

*Table 4.* Cross-domain generalization performance on the real-world FlareReal600 dataset.

| Model | Dataset | PSNR | G-PSNR | LPIPS |
|---|---|---|---|---|
| Flare7K | Full | 19.585 | 0.592 | 0.261 |
| Flare7Kpp | Full | 19.890 | 0.599 | **0.257** |
| SFHformer | Full | 20.117 | 0.602 | 0.266 |
| **Ours** | 8K + 8K | **20.368** | **0.604** | 0.260 |

*Table 5.* Effect of incorporating unlabeled data under full supervision (using Uformer).

| Dataset | PSNR | SSIM | LPIPS | G-PSNR | S-PSNR |
|---|---|---|---|---|---|
| 8K Labled | 27.392 | 0.890 | 0.0455 | 23.982 | 21.506 |
| 8K + 8K | 27.467 | 0.896 | 0.0458 | 24.046 | 22.751 |
| Full Labled | 27.633 | 0.894 | 0.0428 | 23.949 | 22.603 |
| Full + 8K | 28.045 | 0.906 | 0.0414 | 24.589 | 23.436 |

*Table 6.* Ablation study on strong augmentations applied to unlabeled data.

| CJ | RG | GB | PSNR | SSIM | LPIPS | G-PSNR | S-PSNR |
|---|---|---|---|---|---|---|---|
| × | × | × | 26.796 | 0.888 | 0.0531 | 23.384 | 21.684 |
| ✓ | × | × | 27.342 | 0.892 | 0.0511 | 23.976 | 22.361 |
| × | ✓ | × | 27.349 | 0.895 | 0.0508 | 23.942 | 22.514 |
| × | × | ✓ | 27.159 | 0.895 | 0.0521 | 23.612 | 22.360 |
| ✓ | ✓ | ✓ | **27.620** | **0.897** | **0.0493** | **24.276** | **22.686** |

CJ: Color Jitter, RG: Random Grayscale, GB: Gaussian Blur.

additionally conduct cross-domain evaluation on the real-world FlareReal600 dataset. (Table 4) Although Flare7K++ obtains a slightly better LPIPS score, our method provides a better balance between pixel fidelity and structural restoration.

**Subjective evaluation.** Shown in Figure 4 on the Flare7K++ test set, fully supervised models trained with limited labeled data often leave noticeable residual flare artifacts. In contrast, our method achieves more effective flare suppression while better preserving background structures and local details, leading to visually more balanced and natural results.

Moreover, when compared with fully supervised models trained on the complete paired dataset, our results remain visually competitive, indicating that the proposed framework can narrow the performance gap caused by reduced supervision. Due to space limitations, additional qualitative comparisons on FlareX and FlareReal600 are provided in

the supplementary material.

### 4.4. Ablation Study

**Effect of core components.** Table 2 evaluates the impact of the dependable repository (DR) and the flare-aware contrastive loss $\mathcal{L}_{cr}$. Removing either component leads to consistent performance degradation. In particular, w/o DR produces unstable pseudo supervision and visible residual

*Table 7.* Ablation study isolating the contribution of NR-IQA-based quality filtering in the dependable repository.

| DR Strategy | PSNR | SSIM | LPIPS | G-PSNR | S-PSNR |
|---|---|---|---|---|---|
| None | 26.746 | 0.885 | 0.0533 | 23.427 | 21.658 |
| Mom. | 26.982 | 0.891 | 0.0518 | 23.502 | 22.065 |
| Mom. + QG | **27.620** | **0.897** | **0.0493** | **24.276** | **22.686** |

flare artifacts, while w/o $\mathcal{L}_{cr}$ results in blurred structures and halo-like distortions. The full model achieves the best quantitative results and generates visually cleaner images with improved structure preservation, indicating that DR and contrastive regularization are complementary (Shown in Figure 5).

**Contribution of NR-IQA quality filtering.** We further analyze the effect of quality filtering in the dependable repository. As shown in Table 7, momentum-based pseudo-label updating already improves the baseline, while adding the NR-IQA-based quality gate brings further gains. This shows that momentum updating helps stabilize pseudo supervision, and quality filtering further prevents unreliable teacher predictions from being accumulated in the repository. Here, "None" denotes removing the dependable repository, "Mom." denotes momentum-based pseudo-label updating without quality filtering, and "Mom.+QG" denotes the full strategy with both momentum updating and NR-IQA-based quality gating.

**Benefit of unlabeled data beyond fully supervised training.** As shown in Table 5 and Figure 1, adding 8K unlabeled images to fully supervised training consistently improves both objective metrics and visual quality. Compared with the purely supervised Uformer, the augmented model better suppresses large-scale flare and preserves fine textures in challenging regions. This suggests that reliable pseudo supervision provides additional regularization and scene diversity beyond paired data, even when full supervision is available.

**Impact of strong augmentations on unlabeled data.** Table 6 analyzes the role of strong augmentations for unlabeled data. Without strong augmentation, the model exhibits limited gains and tends to overfit to pseudo targets. Removing any single augmentation degrades performance, indicating that each component contributes complementary regularization effects, while using all three yields the best results. From Figure 6, the full-augmentation setting produces more robust flare removal across varying illumination and color conditions, with fewer residual artifacts and better structural preservation, confirming that diverse perturbations improve generalization.

## 5. Conclusion

In this paper, we present a semi-supervised framework for flare removal that effectively exploits unlabeled data under limited paired supervision. We introduce a reliable pseudo-label repository that progressively refines pseudo targets through quality-aware updating and persistence, providing stable supervision for unlabeled samples. In addition, we propose a flare-aware contrastive regularization that explicitly encourages discriminative representation learning by separating flare-contaminated inputs from reliable pseudo targets at the patch level. Together, these components significantly improve training stability and robustness without introducing additional complexity at inference time. Extensive experiments on multiple flare benchmarks demonstrate that our method achieves performance comparable to fully supervised state-of-the-art models while requiring substantially fewer labeled samples. This work provides a practical and extensible direction for data-efficient flare removal and related image restoration tasks.

## Acknowledgments

This work was supported by the Natural Science Foundation of China (62202347).

## Impact Statement

This paper presents work whose goal is to advance the field of Machine Learning. There are many potential societal consequences of our work, none which we feel must be specifically highlighted here.

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

# A. Detailed Architecture of RaLiFormer

This section details the generator architecture of RaLi-Former used in both student and teacher networks. RaLi-Former adopts a symmetric encoder-decoder structure with multi-scale feature modeling and skip connections, enabling effective suppression of spatially extended flare artifacts while preserving background structures.

## A.1. Overall Architecture

Given an input image $\mathbf{I} \in \mathbb{R}^{H \times W \times 3}$, RaLiFormer first maps it into an embedding space:

$$X_0 = f_{\text{in}}(\mathbf{I}), \tag{10}$$

where $f_{\text{in}}(\cdot)$ denotes a convolutional projection. The features are processed through encoder stages, a bottleneck, and mirrored decoder stages:

$$X_{s+1} = \mathcal{E}_s(X_s), \quad X_{s-1} = \mathcal{D}_s(X_s) + X_s^{\text{skip}}, \tag{11}$$

where $\mathcal{E}_s(\cdot)$ and $\mathcal{D}_s(\cdot)$ denote encoder and decoder blocks composed of stacked PSCBlocks, and $X_s^{\text{skip}}$ represents skip connections.

## A.2. Parallel Spatial-Channel Block

Each PSCBlock jointly models spatial dependencies and channel-wise adaptation. Given an intermediate feature map $X \in \mathbb{R}^{H \times W \times C}$, the block computes:

$$X' = X + \mathcal{A}(X) + \mathcal{G}(X), \tag{12}$$

where $\mathcal{A}(\cdot)$ is a rank-enhanced attention operator and $\mathcal{G}(\cdot)$ is a channel attention function.

## A.3. Channel Attention Block

The channel attention branch $\mathcal{G}(\cdot)$ is implemented using a Channel Attention Block (CAB), which adaptively re-weights feature channels based on their global responses. Given an input feature map $X \in \mathbb{R}^{H \times W \times C}$, spatial information is first aggregated by global average pooling:

$$z_c = \frac{1}{HW} \sum_{i=1}^{H} \sum_{j=1}^{W} X_{i,j,c}, \tag{13}$$

yielding a compact channel descriptor $z \in \mathbb{R}^C$.

This descriptor is then passed through a lightweight gating function consisting of two fully connected layers with a non-linear activation:

$$s = \sigma\left(W_2 \delta\left(W_1 z\right)\right), \tag{14}$$

where $\delta(\cdot)$ denotes ReLU and $\sigma(\cdot)$ denotes the sigmoid function. The input features are finally recalibrated via channel-wise multiplication:

$$\mathcal{G}(X) = X \odot s. \tag{15}$$

## A.4. Rank-Enhanced Linear Attention

The spatial attention branch is implemented using Rank-Enhanced Linear Attention (ReLinA), which captures long-range spatial dependencies with linear complexity. Given an input feature map $X \in \mathbb{R}^{H \times W \times C}$, query, key, and value tensors are obtained through linear projections:

$$\mathbf{Q} = XW_Q, \quad \mathbf{K} = XW_K, \quad \mathbf{V} = XW_V. \tag{16}$$

A positive feature mapping is applied to stabilize the linear attention computation:

$$\mathbf{Q}' = \phi(\mathbf{Q}), \quad \mathbf{K}' = \phi(\mathbf{K}), \quad \phi(x) = 1 + \text{ELU}(x). \tag{17}$$

The attention output is computed as:

$$\mathcal{A}(X) = \frac{\mathbf{Q}'(\mathbf{K}'^{\top}\mathbf{V})}{\mathbf{Q}'\left(\sum_j \mathbf{K}'_j\right) + \varepsilon}, \tag{18}$$

where $\varepsilon$ is a small constant for numerical stability.

To mitigate the rank deficiency commonly observed in linear attention, ReLinA refines the value features using spatial-domain enhancement:

$$\tilde{\mathbf{V}} = \mathbf{V} + \text{DWConv}(\mathbf{V}), \tag{19}$$

where $\text{DWConv}(\cdot)$ denotes depth-wise convolution. The refined values are then used in attention computation, allowing ReLinA to incorporate both global context and local structural cues.

## A.5. Directionally-Enhanced Feed-Forward Network

Following the dual-attention aggregation, each PSCBlock employs a Directionally-Enhanced Spatial Module (DESM) as its feed-forward component. The design of DESM is motivated by the observation that flare artifacts often exhibit strong directional characteristics, such as streaks, rays, and anisotropic halo structures, which are not explicitly modeled by conventional channel-wise feed-forward networks.

Given an input feature map $X \in \mathbb{R}^{H \times W \times C}$, DESM first expands the feature dimension and splits it into two complementary branches:

$$[X_1, X_2] = \text{Split}\left(\mathcal{E}(X)\right), \tag{20}$$

where $\mathcal{E}(\cdot)$ denotes a channel expansion operation. The two branches are designed to capture different aspects of spatial representation.

The first branch focuses on spatial modeling and applies lightweight directional depth-wise convolutions to enhance responses along dominant orientations. The second branch serves as a gating pathway that adaptively modulates spatial

features. The outputs of the two branches are combined through element-wise interaction and projected back to the original embedding space:

$$\mathcal{F}(X) = \mathcal{P}\big(\mathcal{D}(X_1) \odot X_2\big), \qquad (21)$$

where $\mathcal{D}(\cdot)$ denotes directional depth-wise convolution, $\odot$ represents element-wise multiplication, and $\mathcal{P}(\cdot)$ is a linear projection.

## B. Implementation Details

### B.1. Dataset Construction and Data Augmentation

During training, we adopt the same data synthesis and augmentation strategy as Flare7K++ to ensure a fair and consistent comparison. Both flare images and background images are first subjected to inverse gamma correction with $\gamma \sim \mathcal{U}(1.8, 2.2)$ to restore linear brightness. Subsequently, flare images undergo a series of random geometric transformations, including rotation $\mathcal{U}(0, 2\pi)$, translation $\mathcal{U}(-300, 300)$, shear $\mathcal{U}(-20°, 20°)$, and scaling $\mathcal{U}(0.8, 1.5)$, followed by Gaussian blur with $\sigma \sim \mathcal{U}(0.1, 3.0)$ and random horizontal flipping.

To simulate global illumination effects introduced by lens flares, a random intensity offset sampled from $\mathcal{U}(-0.02, 0.02)$ and color jitter are applied to the flare images. For background images, RGB channels are scaled by a random factor $\mathcal{U}(0.5, 1.2)$, and additive Gaussian noise with variance sampled from a scaled chi-square distribution $\sigma^2 \sim 0.01\chi^2$ is introduced. The enhanced flare image is then composited with the background image, and the resulting flare-corrupted image is clipped to the range $[0, 1]$ and used as the network input. The corresponding flare-free ground truth is obtained by subtracting the flare component from the synthesized image.

Overall, the entire data construction pipeline strictly follows the protocol of Flare7K++, ensuring that performance gains originate from the proposed model and training strategy rather than differences in data synthesis.

### B.2. Training Configuration and Optimization

All experiments are conducted using cropped image patches of size $512 \times 512$. The student and teacher networks share the same generator architecture, while the teacher parameters are updated using an exponential moving average (EMA) of the student parameters with a momentum coefficient of 0.999. The model is trained for a total of 40 epochs using the Adam optimizer, with $\beta_1 = 0.9$ and $\beta_2 = 0.999$, and a cosine decay schedule is applied after the warm-up stage.

To stabilize early training, a linear warm-up strategy is adopted for the learning rate:

$$\eta(t) = \eta_0 \cdot \frac{t}{N_{\text{warm}}}, \quad t \leq N_{\text{warm}}, \qquad (22)$$

where $t$ denotes the current iteration and $N_{\text{warm}}$ is the total number of warm-up iterations. After warm-up, the learning rate follows the predefined decay schedule.

To further enhance training diversity and improve generalization, we additionally employ a Mixup strategy on the fully supervised training samples starting from epoch 10. By linearly interpolating pairs of labeled images and their corresponding ground truths, Mixup effectively increases the complexity of the training distribution and encourages the model to learn smoother and more robust representations. This strategy is applied exclusively to the supervised subset and does not affect the unlabeled data or the semi-supervised consistency learning process. For supervised samples $(x_i, y_i)$ and $(x_j, y_j)$, Mixup generates interpolated inputs and targets as:

$$\tilde{x} = \lambda x_i + (1 - \lambda)x_j, \quad \tilde{y} = \lambda y_i + (1 - \lambda)y_j, \qquad (23)$$

with $\lambda \sim \text{Beta}(\alpha, \alpha)$.

For semi-supervised learning, the unsupervised consistency loss is gradually introduced using a ramp-up weighting strategy to avoid noisy supervision in early epochs. The overall training objective combines supervised and unsupervised losses, and gradient accumulation is employed to maintain stable optimization under limited GPU memory. During inference, only the student network is used, and no additional architectural components or computational overhead are introduced.

## C. Subjective evaluation

The qualitative comparison results on the FlareX and Flare-Real600 test sets are shown below. It can be observed that existing methods often leave residual flare artifacts, diffuse halos, or introduce noticeable over-smoothing, particularly in regions with strong illumination and complex structures.

In contrast, our method achieves more effective flare suppression while better preserving background textures and fine structural details. On real-world scenes from Flare-Real600, the proposed model exhibits stronger robustness to diverse and spatially extended flare patterns, removing halos without causing color shifts or structural degradation. These visual comparisons further demonstrate the superiority and generalization capability of our approach under both synthetic and real flare conditions. Moreover, on the flare-corrupted test images from Flare7K++, our approach consistently reduces both streak and halo artifacts. These visual comparisons further demonstrate the superiority and generalization capability of our approach under both synthetic and real flare conditions.

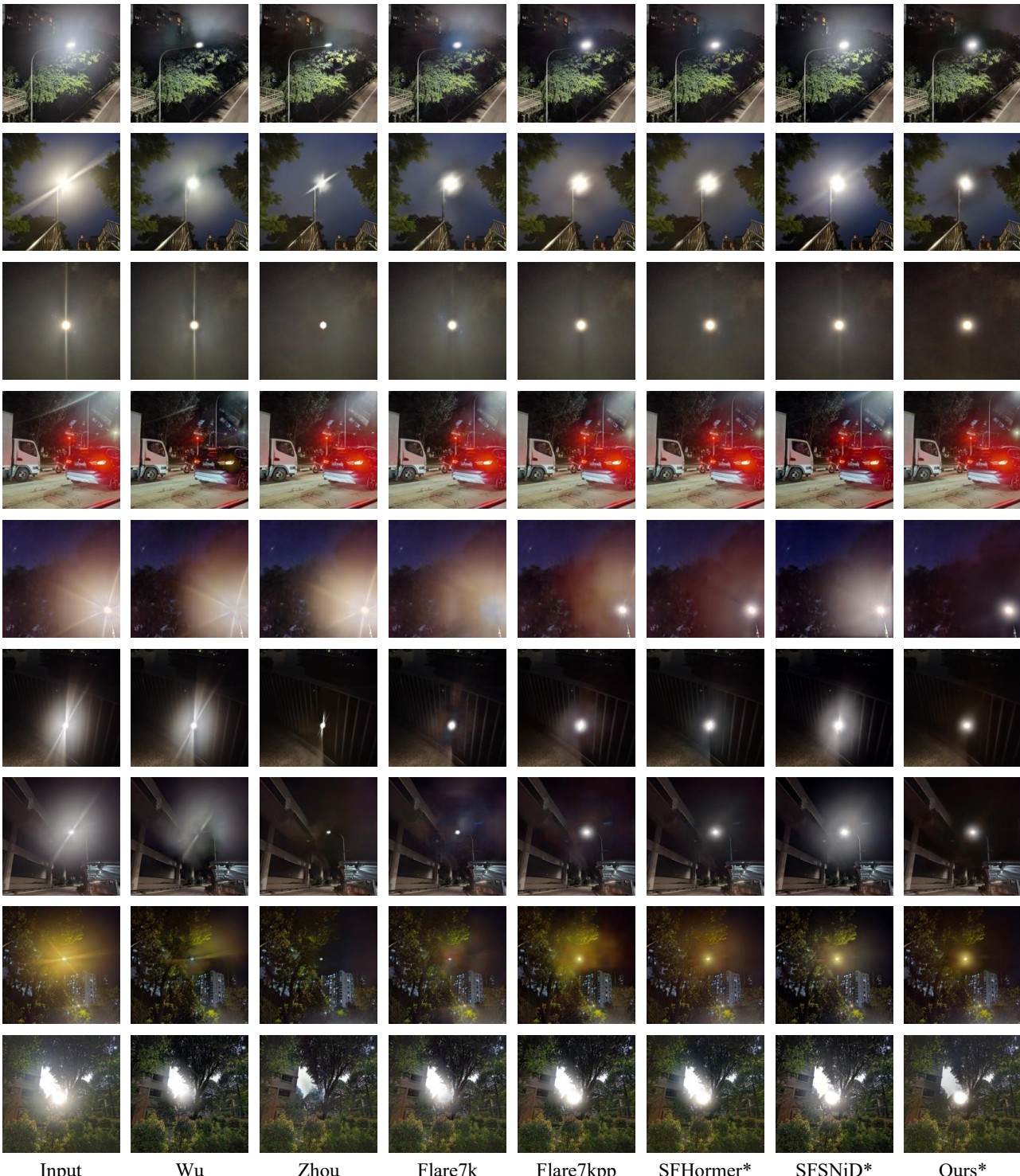

| Input | Wu | Zhou | Flare7k | Flare7kpp | SFHormer* | SFSNiD* | Ours* |

*Figure 7.* Visual comparison of flare removal on flare-corrupted test. * indicates models trained with 8K labeled and 8K unlabeled data.

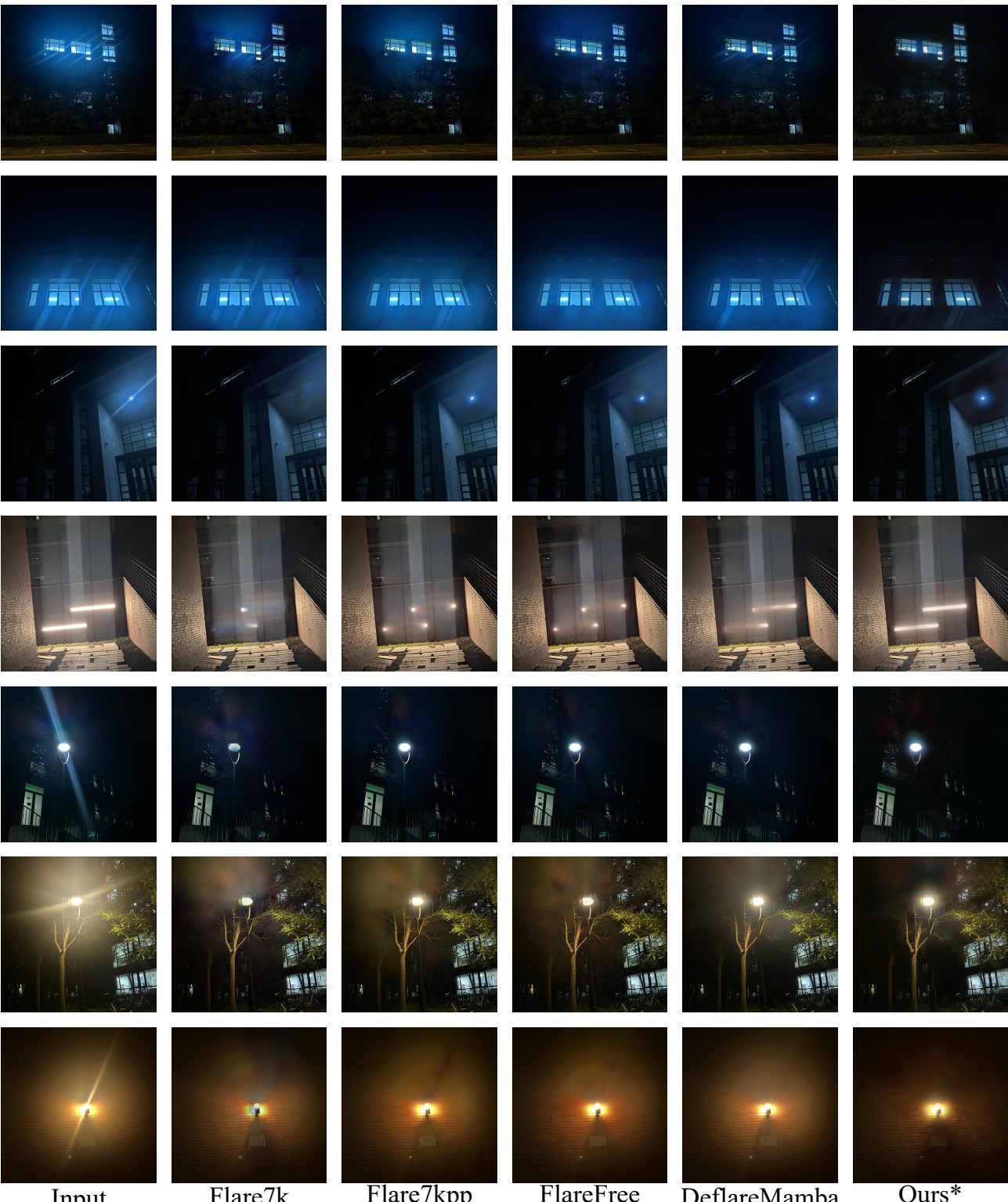

| Input | Flare7k | Flare7kpp | FlareFree | DeflareMamba | Ours* |

*Figure 8.* Visual comparison of flare removal on FlareX test. * indicates models trained with 8K labeled and 8K unlabeled data.

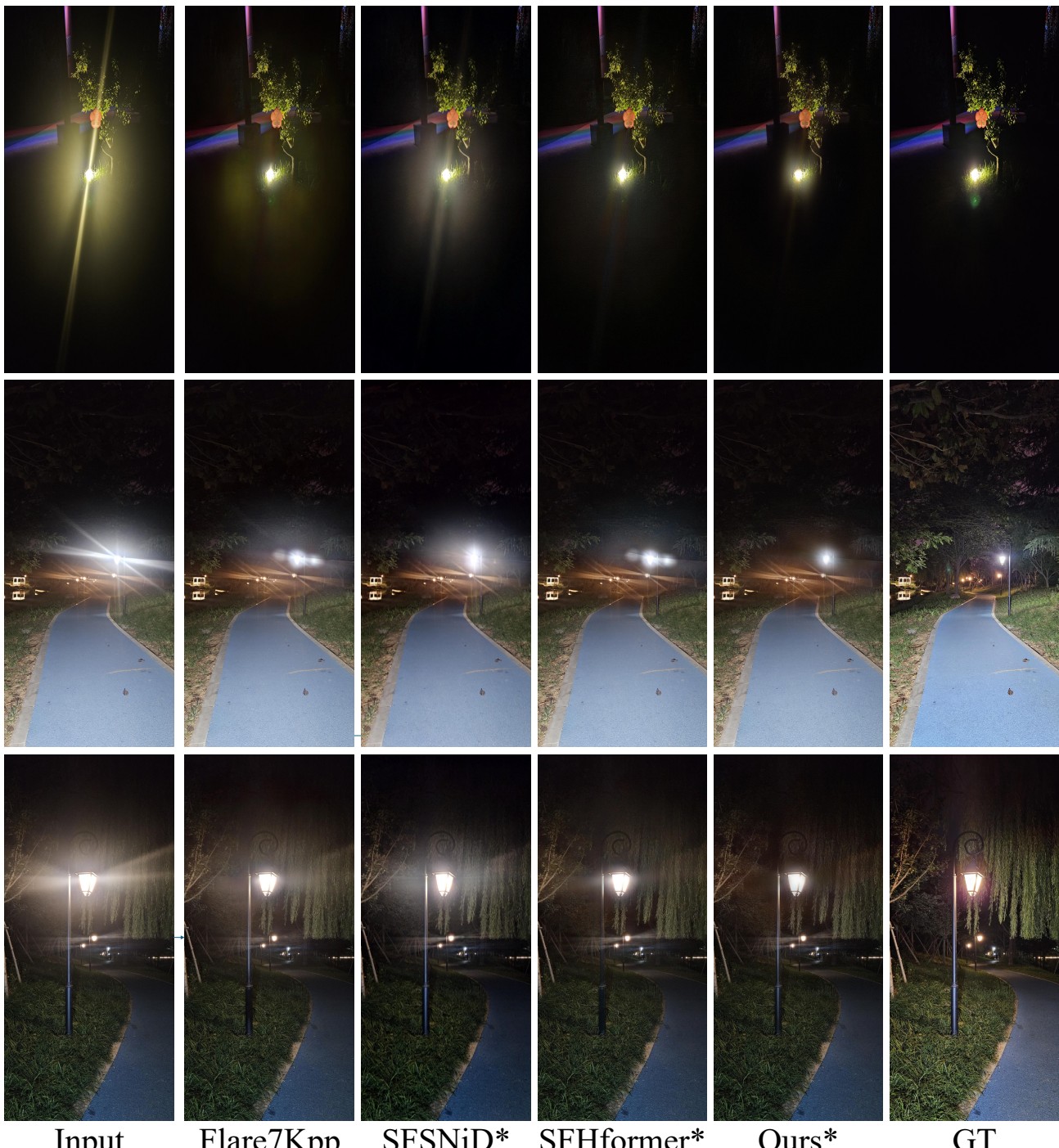

Input     Flare7Kpp     SFSNiD*     SFHformer*     Ours*     GT

*Figure 9.* Visual comparison of flare removal on FlareReal600 test. * indicates models trained with 8K labeled and 8K unlabeled data.

