# OpenReview forum: "Semi-LAR: Semi-supervised Contrastive Learning with Linear Attention for Removal of Nighttime Flares"
_ICML.cc/2026/Conference — ICML 2026 regular_

### Official Review · Reviewer_6YvR · 2026-03-05

**Soundness:** 3
**Presentation:** 3
**Significance:** 3
**Originality:** 3
**Overall Recommendation:** 5
**Confidence:** 5

**Summary:**

This paper introduces Semi-LAR, a semi-supervised framework for nighttime flare removal while reduces the dependency on large-scale paired datasets. The framework addresses the instability of pseudo-labels and feature entanglement through an adaptive pseudo-label repository guided by NR-IQA and momentum updates, and a flare-aware contrastive loss. By treating flare-contaminated inputs as structured negatives, the method leverages unlabeled with fewer labeled samples approaching fully-supervised performance.

**Compliance With Llm Reviewing Policy:**

Affirmed.

**Final Justification:**

The author's rebuttal addressed my concerns, and I raise my score to accept 5 in the final decision.

**Key Questions For Authors:**

See weakness.

**Limitations:**

Yes

**Strengths And Weaknesses:**

Strengths
1、The paper tackles a fundamental problem for low level vison, strong dependence on synthetic paired data, which restricts network generalization. The proposed semi-supervised framework can use real-world unlabeled flare images, which is well motivated for real world applications, when ground truth is difficult to obtain.
2、The proposed reliability-aware pseudo-label is designed to be high-quality and smooth. The quality-guided repository, combined with NR-IQA-based filtering and momentum refinement, provides a principled way to stabilize pseudo-supervision and mitigate error accumulation.
Weaknesses
1、While the dependable repository is a key component of the framework, the specific contribution of the NR-IQA (MUSIQ) based quality filtering is not sufficiently validated. It remains unclear whether the performance improvements stem merely from the repository's persistence mechanism (momentum updates) or from the explicit quality control. An ablation study via isolating the MUSIQ-based gating is needed.
2、Though the proposed method use limited and unlabeled data, its quantitative performance gains over strong supervision baselines are relatively marginal. For better justification, the cross dataset generalization evaluation is recommended.

---

> ### Author Rebuttal · Authors · 2026-03-30
>
> We sincerely thank Reviewer 6YvR for recognizing the fundamental and well motivation of our work, the sound design of our reliability-aware pseudo-label repository to achieve approaching fully-supervised performance. We address your insightful comments point to point:
>
> ## **1. Isolating the Contribution of NR-IQA (MUSIQ) Quality Filtering:**
>
> We completely agree with your assessment. In our original manuscript, the "w/o DR" ablation simultaneously removed both the momentum updates and the quality gating. To rigorously validate the specific contribution of the MUSIQ-based filtering, we conducted the exact ablation study as you recommended.
>
> We trained a variant of our model where the pseudo-labels in the repository are updated using momentum at every step, as the **MUSIQ-based quality gating is entirely disabled**. The results on the Flare7K++ test set are shown below:
>
> **Table: Ablation study isolating the specific contribution of MUSIQ-based quality gating in the Dependable Repository.**
>
> | Model Variant                  | PSNR ↑     | SSIM ↑    | LPIPS ↓    | G-PSNR ↑   | S-PSNR ↑   |
> | :----------------------------- | :--------- | :-------- | :--------- | :--------- | :--------- |
> | w/o DR (No Momentum, No MUSIQ) | 26.746     | 0.885     | 0.0533     | 23.427     | 21.658     |
> | _Momentum Only (w/o MUSIQ)_    | _26.982_   | _0.891_   | _0.0518_   | _23.502_   | _22.065_   |
> | **Full Model**                 | **27.620** | **0.897** | **0.0493** | **24.276** | **22.686** |
>
> _Analysis:_ As the table demonstrates, relying solely on momentum updates without quality control leads to a performance drop. Without MUSIQ to act as a strict gatekeeper, the repository inevitably absorbs flawed teacher predictions, leading to error accumulation over time. The explicit quality control provided by MUSIQ is therefore indispensable for stabilizing the pseudo-supervision. We will include this isolated ablation in the revised manuscript.
>
> ## **2. Cross-Dataset Generalization Evaluation:**
>
> We appreciate your suggestion to evaluate cross-dataset generalization to justify the performance gains. We would like to respectfully direct your attention to **Table 1** of the manuscript, which was specifically designed to address this exact point.
>
> In Table 1 of the manuscript, the models were trained on a partial subset of the Flare7K++ dataset and evaluated on the completely independent **FlareX** benchmark. As the results show, our semi-supervised framework (Ours: 8K + 8K) achieved a PSNR of 19.960 and a G-PSNR of 29.899 on FlareX, explicitly demonstrating that our method maintains highly robust generalization in the cross-dataset, even outperforming several fully-supervised models trained on the complete dataset. Another cross domain evaluation on FlareReal600 is conducted as below. Our method achieves the highest PSNR and SSIM on this unseen **FlareReal600** dataset with 0.251 on PSNR gains.
>
> **Table: Cross-domain generalization performance on FlareReal600**
>
> | Method    | PSNR ↑     | SSIM ↑    | LPIPS ↓   |
> | :-------- | :--------- | :-------- | :-------- |
> | Flare7K   | 19.585     | 0.592     | 0.261     |
> | Flare7K++ | 19.890     | 0.599     | **0.257** |
> | SFHformer | 20.117     | 0.602     | 0.266     |
> | **Ours**  | **20.368** | **0.604** | 0.260     |
>
> Furthermore, to strengthen this point visually, we have provided extensive qualitative comparisons in the **Supplementary Material** (e.g., Figure 8 for FlareX and Figure 9 for the real-world FlareReal600 test set). These visual results clearly demonstrate our model's superior ability to generalize and suppress complex flares in diverse real-world scenarios.

---

> > ### Author Rebuttal · Reviewer_6YvR · 2026-04-02
> >
> > The author's rebuttal addressed my concerns, and I will consider raising my score in the final decision.

---

> > > ### Author Response · Authors · 2026-04-07
> > >
> > > Thank you very much for your positive feedback. We are encouraged to hear that our additional experiments effectively resolved your concerns. Thank you again for your valuable time and comments.

---

### Official Review · Reviewer_1gUF · 2026-03-11

**Soundness:** 2
**Presentation:** 3
**Significance:** 3
**Originality:** 3
**Overall Recommendation:** 4
**Confidence:** 3

**Summary:**

The paper proposes Semi-LAR, a semi-supervised framework that uses a teacher model to learn from unlabeled images. It introduces two main components: a pseudo-label repository that filters pseudo labels based on image quality scores, and a flare-aware contrastive loss. Experiments show that the method can achieve competitive results using limited labeled data with unlabeled images

**Compliance With Llm Reviewing Policy:**

Affirmed.

**Final Justification:**

The original setting used synthetic data for training. The new rebuttal experiments which train on real images address my main concern about this semi-supervised method. After reviewing the rebuttal and other reviewer's comment, I raise my score.

**Key Questions For Authors:**

1. How does the method perform when the unlabeled data comes from real flare images rather than the same synthetic generation pipeline?

2. Can the proposed semi-supervised framework be applied to additional backbones besides Uformer? It would be great if the authors could provide more results on this to further show that the proposed method is "model-agnostic".

3. The dependable repository relies heavily on the MUSIQ metric. Could the authors provide a discussion whether MUSIQ specifically correlates with flare removal quality (or provide visual examples of teacher outputs with their corresponding MUSIQ scores)?

**Limitations:**

See Weaknesses.

**Strengths And Weaknesses:**

Strengths:

1. The paper addresses a practical problem as collecting paired flare data is difficult in real scenarios.
2. The flare-aware contrastive loss is intuitive for this task and appears to bring great gains in the ablation study.
3. The method outperforms several baselines on metrics such as G-PSNR and S-PSNR, and the visual comparisons shown in the paper are appealing.

Weaknesses:

1. The main contribution of this paper is semi-supervised learning for flare removal. The architectural novelty is very limited. The proposed linear attention model is not fundamentally new.  Since the authors mention that "the proposed framework is model-agnostic", I suggest that the authors focus on evaluations on more backbones.
2. Another key point is that the semi-supervised setting here may not fully reflect real-world scenarios. The unlabeled data are still derived from the same synthetic pipeline in Flare7K++. This raises concerns that the teacher model may already perform well on this in-domain data and therefore questions the effectiveness of the proposed semi-supervised learning.
3. The dependable repository relies heavily on the MUSIQ metric, which is a general image quality metric. It might not be suitable for flare removal, especially whne it is used to filter samples as pseudo ground truth.

---

> ### Author Rebuttal · Authors · 2026-03-30
>
> We thank Reviewer 1gUF for appreciating our practical value and intuitive design. We address your insightful comments below:
>
> ## **1.  Contribution and Additional Backbones:**
>
> Our contribution is a problem-driven architecture tailored to nighttime flares:
>
> From Global Contrastive to Flare-Aware Disentanglement: Unlike contrastive learning with global representations, our strategy operates at the patch level and explicitly treats flare-contaminated regions as "hard negatives", forcing the network to decouple flare patterns from textures.
>
> From Static Filtering to a Dynamic Quality Gate: We utilize NR-IQA not merely as a filter, but as a physical quality gate to break the error-accumulation loop. Dynamically refining pseudo-labels via momentum updates ensures the model learns from progressively reliable targets, relaxing the dependency on pseudo-label accuracy.
>
> Closed-Loop Synergy: contrastive loss and DR form a closed loop. The DR provides the high-quality, reliable targets for the contrastive loss to function effectively, while the contrastive loss equips the student network with robust decoupling capability needed to generate better candidates for the DR.
>
> We conducted experiments on more backbones(**Table X**). Improvements on all 5 evaluation metrices across 3 architectures validates our effectiveness.
>
> **Table X: Quantitative results of different backbones.**
>
> | Backbone    | Supervision Setting           | PSNR ↑     | SSIM ↑    | LPIPS ↓    | G-PSNR ↑   | S-PSNR ↑   |
> | :---------- | :---------------------------- | :--------- | :-------- | :--------- | :--------- | :--------- |
> | U-Net       | Limited-Sup. (8K Labeled)     | 25.098     | 0.856     | 0.0562     | 22.109     | 19.588     |
> | U-Net       | **Semi-Sup. (Ours: 8K + 8K)** | **25.531** | **0.874** | **0.0549** | **22.434** | **21.228** |
> | NAFNet      | Limited-Sup. (8K Labeled)     | 26.553     | 0.863     | 0.0529     | 23.437     | 21.321     |
> | NAFNet      | **Semi-Sup. (Ours: 8K + 8K)** | **27.112** | **0.892** | **0.0505** | **24.156** | **22.001** |
> | Restormer\* | Limited-Sup. (8K Labeled)     | 27.277     | 0.889     | **0.0497** | 23.970     | 21.557     |
> | Restormer\* | **Semi-Sup. (Ours: 8K + 8K)** | **27.431** | **0.898** | 0.0502     | **24.145** | **22.025** |
>
> ## **2. Unlabeled Data and Generalization to Real-World Flares:**
>
>  We can use real world unlabeled images for training, but existing SOTAs all adopt Flare7K++ synthesis pipeline. Thus, we also follow this to ensure a strictly fair comparison while using less data. Our framework is efficient with only $1/3$ dataset as 8K labeled and 8K unlabeled data, while achieves performance comparable to full supervision.
>
> We address the concern of possible in-domain overfitting from two aspects. Our architecture effectiveness is shown in Table. 2 of manuscript, as any module removal cause PSNR drop between 0.909 and 1.11. The unlabeled data effectiveness  is validated in Table 4, semi-supervised learning on full dataset (Full + 8K Unlabeled) gains 0.412 PSNR improvement, compared to supervised training on same dataset (Full Labeled) with PSNR of 27.633.
>
> **Generalization to Real-World Flares:** To show our generalization on real world flares, we evaluated directly on the unseen **FlareReal600** dataset in Table Y below and FlareX dataset in Table 1 of manuscript.
>
> **Table Y: Cross-domain generalization performance on the real-world FlareReal600 dataset.**
>
> | Method    | PSNR ↑     | SSIM ↑    | LPIPS ↓   |
> | :-------- | :--------- | :-------- | :-------- |
> | Flare7K   | 19.585     | 0.592     | 0.261     |
> | Flare7K++ | 19.890     | 0.599     | **0.257** |
> | SFHformer | 20.117     | 0.602     | 0.266     |
> | **Ours**  | **20.368** | **0.604** | 0.260     |
>
> Our method ranks first on FlareReal600 with 0.251 on PSNR gains, and 0.464 PSNR improvement on FlareX on Table 1 of manuscript. Visual comparisons are also given in the supplementary material (Fig. 8-9).
>
> ## **3. Dependence on the MUSIQ Metric and Ablation Study:**
>
> Although MUSIQ is a general-purpose metric, its role in our Dependable Repository (DR) is a constrained perceptual prior rather than a decision-maker. In Sec. 3.2 and Alg. 1, candidate pseudo-labels must first pass Physical Bounding and Validity Gating before being evaluated by MUSIQ, which ensures the physical plausibility before assessing perceptual quality.
>
> We also conducted an experiment as **Table Z** and shows the efficiency of DR components:
>
> **Table Z: Ablation study on Dependable Repository.**
>
> | Model Variant   | PSNR ↑     | SSIM ↑    | LPIPS ↓    | G-PSNR ↑   | S-PSNR ↑   |
> | :-------------- | :--------- | :-------- | :--------- | :--------- | :--------- |
> | w/o DR (No Momentum, No MUSIQ) | 26.746     | 0.885     | 0.0533     | 23.427     | 21.658     |
> | _Momentum Only (w/o MUSIQ)_    | _26.982_   | _0.891_   | _0.0518_   | _23.502_   | _22.065_   |
> | **Full Model**                 | **27.620** | **0.897** | **0.0493** | **24.276** | **22.686** |

---

> > ### Author Rebuttal · Reviewer_1gUF · 2026-04-03
> >
> > Thank you for the rebuttal. Could the authors further explain W2? My question is not about generalization to real-world flares. Rather, since the method proposes a semi-supervised approach that can leverage unlabeled real-world images during training, why is the training data still derived from the same synthetic pipeline and based only on synthetic data? I do not think fairness of comparison is a sufficient reason here, because the proposed semi-supervised method seems likely to become much more robust when scaling up with real training data, and that is in fact one of its main strengths. Could the authors share more insights on this?

---

> > > ### Author Response · Authors · 2026-04-07
> > >
> > > We appreciate the reviewer's interest in the potential of a semi-supervised approach to real-world data training.
> > >
> > > To fully demonstrate that our method's effectiveness from its semi-supervised mechanisms, we added new experiments in the last 4 days for real-world data training only by excluding all synthetic data, preventing it from being "derived from the same synthetic pipeline and based only on synthetic data."
> > >
> > > ## **1. Real-world training data without synthetic pipeline**
> > >
> > > We conducted a semi-supervised training on a small FlareReal600[1] dataset with 600 paired data, which is introduced as "the first real paired nighttime flare removal dataset," much smaller than 24K Flare7kpp[2]. In this setup, the labeled training set consisted solely of the 600 real-world images with ground truth (GT) from FlareReal600. For our semi-supervised approach without any synthetic pipeline, we combined this with a set of 600 real unlabeled nighttime images without GT (collected from
> > >
> > > https://bing.com/images/search?q=Road+Street+Lamp+Night&form=RESTAB&first=1&cw=1513&ch=731,
> > >
> > > https://bing.com/images/search?q=Street+Lamp+Post+at+Night&form=RESTAB&first=1,
> > >
> > > https://bing.com/images/search?q=Street+Lamp+Night+Road&form=RESTAB&first=1) for training.
> > >
> > > The baseline adopted here is Ufromer used in Flare7kpp. The trained models were then tested on the FlareReal600 and FlareX test sets. This experiment confirms that our Semi-LAR, even with a very small labeled set (600 images), our DR and $\mathcal{L}_{cr}$ effectively harvest reliable physical priors from unconstrained real-world data to obtain a +0.11 improvement in SSIM on the unseen FlareX dataset.
> > >
> > > **Table 1: Performance comparison under FlareReal600 Training.**
> > > | Method | Training Data | ┃ | FlareReal600 (PSNR) | FlareReal600 (SSIM) | FlareReal600 (LPIPS) | ┃ | FlareX (PSNR) | FlareX (SSIM) | FlareX (LPIPS) |
> > > |-|-|-|-|-|-|-|-|-|-|
> > > | Fully-supervised Baseline | 600 Real | ┃ | 17.692 | 0.447 | 0.6607 | ┃ | 24.308 | 0.483 | 0.1552 |
> > > | Ours (Semi-LAR) | 600 Real + 600 Real Unlabeled | ┃ | **18.582** | **0.569** | **0.2983** | ┃ | **24.523** | **0.599** | 0.1673 |
> > >
> > > ## **2. Real-world training data with the same synthetic pipeline**
> > >
> > >  Flare7Kpp is not a totally synthetic nighttime flare removal dataset, with a very small subset as “962 real-captured flare images (Flare-R)” (described in the abstract of [2]). To validate real-world data training with the most SOTAs adopted Flare7Kpp synthetic pipeline, we utilized only Flare-R as our labeled training set, which incorporates an additional 1K unlabeled real-world nighttime images without GT (same website source as above). This ensures the reproducibility of our training setup for other researchers.
> > >
> > > Both models were evaluated directly on the real test data subset of Flare7Kpp, as well as the unseen FlareReal600 and FlareX test sets. The significant performance gain (e.g., a +0.66 dB improvement in PSNR on the FlareX test set) proves that our method's high potential for real-world generalization.
> > >
> > > **Table 2: Performance comparison under Flare-R data training.**
> > > | Method | Training Data | ┃ | Flare7Kpp (PSNR) | Flare7Kpp (SSIM) | Flare7Kpp (LPIPS) | ┃ | FlareReal600 (PSNR) | FlareReal600 (SSIM) | FlareReal600 (LPIPS) | ┃ | FlareX (PSNR) | FlareX (SSIM) | FlareX (LPIPS) |
> > > |-|-|-|-|-|-|-|-|-|-|-|-|-|-|
> > > | Fully-supervised Baseline | 962 Real (Flare-R) | ┃ | 25.011 | 0.844 | 0.0640 | ┃ | 19.412 | 0.583 | 0.2915 | ┃ | 23.758 | 0.563 | 0.1579 |
> > > | Ours (Semi-LAR) | 962 Real (Flare-R) + 1K Real Unlabeled | ┃ | **25.701** | **0.876** | **0.0616** | ┃ | **19.758** | **0.595** | **0.2712** | ┃ | **24.418** | **0.565** | **0.1488** |
> > >
> > > Though restricted by the available real-world training data (<1k), our Semi-LAR has better and more robust performance than SOTAs. Thus, as the reviewer indicates, our method will “become much more robust when scaling up with real training data."
> > >
> > > - [1] FlareReal600: https://github.com/Zdafeng/FlareReal
> > > - [2] Flare7K++: Mixing Synthetic and Real Datasets for Nighttime Flare Removal and Beyond, TPAMI, 2024

---

### Official Review · Reviewer_DLdT · 2026-03-12

**Soundness:** 2
**Presentation:** 3
**Significance:** 4
**Originality:** 3
**Overall Recommendation:** 5
**Confidence:** 3

**Summary:**

The authors propose a semi-supervised flare removal framework built on reliable pseudo-supervision with adaptive pseudo-labeling, designed to progressively refine pseudo-supervision in a fully model-agnostic setting.

**Compliance With Llm Reviewing Policy:**

Affirmed.

**Final Justification:**

After the rebuttal my main concerns were addressed, so I raised the final decision to Accept.

**Key Questions For Authors:**

How do the authors qualitatively discuss the contribution of their model-agnostic framework?

**Limitations:**

The main limitations are primarily related to the presentation of the paper. In particular, there is a lack of clarity in the related work section, which could otherwise strengthen the motivation of the study, as well as in the methodology section, where the model workflow could be described more clearly.

**Strengths And Weaknesses:**

Strengths:
The paper is well structured. The motivation is clear and well presented. The experimental pipeline is well designed, as relevant competing
methods have been benchmarked against the proposed framework across different datasets. The authors also include qualitative examples illustrating the application of their method. Lastly, relevant ablation studies have been conducted, quantitatively confirming the effectiveness of the proposed methodology.

Weaknesses:
There are some concerns regarding the technical soundness of the work. The related work section appears quite fragmented. Given the wide range of models included, it is not entirely clear how the authors intervene to address key research challenges. Additionally, the model section is difficult to follow, and the overall workflow of the model is not easy to understand. Furthermore, some tables and/or figures in the main paper are not explicitly referenced, suggesting either referencing mistakes or that these elements are not essential to the paper.

---

> ### Author Rebuttal · Authors · 2026-03-30
>
> We sincerely thank Reviewer DLdT for acknowledging our "well-presented" motivation, "well-designed" experimental pipeline, and comprehensive benchmarking. With your encouragement, we will thoroughly proofread to fix all unreferenced elements and improve the Related Work in the final version.
>
> Below, we address your key concerns regarding regarding the technical soundness and workflow, related work introduction to our motivation and the element contribution of our framework:
>
> ## **1. Technical Soundness and Workflow:**
> Our core innovation is a problem-driven architecture tailored to nighttime flares with a close-up workflow:
>
> **From Global Contrastive to Flare-Aware Disentanglement**: Unlike standard contrastive learning with global representations, our strategy operates at the patch level and explicitly treats flare-contaminated regions as "hard negatives", forcing the network to decouple flare patterns from underlying textures.
>
> **From Static Filtering to a Dynamic Quality Gate (NR-IQA Guided DR)**: We utilize NR-IQA not merely as a filter, but as a physical quality gate to break the error-accumulation loop. Dynamically refining pseudo-labels via momentum updates ensures the model learns from progressively reliable targets, relaxing the dependency on pseudo-label accuracy.
>
> **Closed-Loop Workflow**: contrastive loss and DR form a closed loop. The DR provides the high-quality, reliable targets for the contrastive loss to function effectively, while the contrastive loss equips the student network with robust decoupling capability needed to generate better candidates for the DR.
>
> Table 2 validates this as removing either component drops PSNR from 27.620 to ~26.7.
>
> ## **2. Related Work to Our Methodology:**
>
> We emphasize the specific bottlenecks that motivated our proposed framework in three folds:
>
> Heavy Dependency on Paired Data. While recent works like Transformer-based and frequency domain models or incorporating additional priors, their high reliance on paired data all restrict their model generalization. We introduce a semi-supervised paradigm that enables the model to leverage diverse and unlabeled data which may contain real-world unseen distributions.
>
> Error Accumulation in Semi-Supervision. Standard semi-supervised methods often suffer from confirmation bias. We design an Adaptive Pseudo-label Repository (DR) guided by NR-IQA. Unlike simple filtering, our DR acts as a perceptual-physical gate to ensure that only reliably improved pseudo-labels guide the training, effectively breaking the error-accumulation loop.
>
> Spatial Entanglement of Flares. Existing teacher–student learning methods often struggle to decouple flare streaks from background textures, as flare varies in local spatial location. We propose a Flare-aware Contrastive Loss ($\mathcal{L}_{cr}$). By explicitly treating local flare-contaminated patches as hard negatives, we force the network to learn discriminative representations that separate artifacts from scene structures.
>
>
> ## **3.Element Contribution of Our Framework:**
>
> In response to your concern on essential elements,  our module ablation studies are available in Table 2 of manuscript, which validates as removing either component drops PSNR from 27.620 to ~26.7. We qualitatively discuss the contribution of our model-agnostic design as below. Our framework acts as a universal solution with many baselines. To quantitatively support this qualitative claim, we evaluated multiple backbones within our framework:
>
> **Table: Quantitative results of different backbones integrated into our semi-supervised framework (8K Labeled + 8K Unlabeled).**
>
> | Backbone    | Supervision Setting           | PSNR ↑     | SSIM ↑    | LPIPS ↓    | G-PSNR ↑   | S-PSNR ↑   |
> | :---------- | :---------------------------- | :--------- | :-------- | :--------- | :--------- | :--------- |
> | U-Net       | Limited-Sup. (8K Labeled)     | 25.098     | 0.856     | 0.0562     | 22.109     | 19.588     |
> | U-Net       | **Semi-Sup. (Ours: 8K + 8K)** | **25.531** | **0.874** | **0.0549** | **22.434** | **21.228** |
> | NAFNet      | Limited-Sup. (8K Labeled)     | 26.553     | 0.863     | 0.0529     | 23.437     | 21.321     |
> | NAFNet      | **Semi-Sup. (Ours: 8K + 8K)** | **27.112** | **0.892** | **0.0505** | **24.156** | **22.001** |
> | Restormer\* | Limited-Sup. (8K Labeled)     | 27.277     | 0.889     | **0.0497** | 23.970     | 21.557     |
> | Restormer\* | **Semi-Sup. (Ours: 8K + 8K)** | **27.431** | **0.898** | 0.0502     | **24.145** | **22.025** |
>
> This means that our approach can adopt many restoration backbones like a conventional U-Net or a Transformer Uformer. All baseline networks obtain improvement in 5 evaluation metrics in Table above. This validates our approach leverages training efficiency with limited unlabeled data. This observation shows our approach has high potential to relax heavy reliance on fully paired synthetic datasets in flare removal task, which is difficult for ground truth capture.

---

> > ### Author Rebuttal · Reviewer_DLdT · 2026-04-03
> >
> > The authors' rebuttal is convincing with respect to my concerns. Overall, I will raise my score to accept.

---

> > > ### Author Response · Authors · 2026-04-07
> > >
> > > Thank you very much for your positive feedback. We are pleased that our clarifications resolved your concerns. Your insightful feedback has been meaningful in improving our work.

---

### Official Review · Reviewer_kx6P · 2026-03-13

**Soundness:** 2
**Presentation:** 2
**Significance:** 2
**Originality:** 3
**Overall Recommendation:** 3
**Confidence:** 5

**Summary:**

This paper addresses the challenge of nighttime lens flare removal, specifically focusing on the high cost of obtaining large-scale paired datasets. The authors propose a semi-supervised framework, "Semi-LAR," which leverages unlabeled data to improve restoration performance. The framework introduces two primary components: (1) a Dependable Pseudo-label Repository that utilizes NR-IQA and momentum updates to refine pseudo-supervision, and (2) a Flare-aware Contrastive Loss that separates restored features from flare-contaminated inputs at the patch level. Additionally, the paper presents a transformer-based generator called RaLiFormer. Experiments on multiple benchmarks (Flare7K++, FlareX, and FlareReal600) show that the method achieves competitive results using significantly fewer labels than fully supervised counterparts.

**Compliance With Llm Reviewing Policy:**

Affirmed.

**Final Justification:**

The authors’ response addressed my concerns; however, the contribution of the paper is somewhat limited.

**Key Questions For Authors:**

See Weaknesses.

**Limitations:**

No, the authors do not include the limitations or the potential societal impact of their work.

**Strengths And Weaknesses:**

Strengths

1. The paper proposes a semi-supervised framework for nighttime flare removal to reduce the reliance on paired synthetic data.
2. The paper presents a novel network for flare removal.

Weaknesses

1. The primary concern is the lack of fundamental technical innovation. The individual components of the proposed framework are well-established in the field of low-level vision: Teacher-Student/EMA, Contrastive Loss, and NR-IQA-based Filtering
2. In the teacher model, a strong augmentation and a weak augmentation are utilized. Is the specific implementation of this augmentation consistent with the explanation in Sec. B.1? If so, the augmentation introduces a synthetic flare simulation model, which could further widen the gap between the simulated flare and real-world flare. This seems to conflict with the objective of using a semi-supervised approach to address real-world scenarios.
3. Has the network in Fig. 2 been replaced with existing restoration networks (such as U-Net, NAFNet[1]) or more specialized deflare networks (such as Flare-level net[2])? Doing so could provide stronger evidence to validate the effectiveness of the proposed framework.
4. It is necessary to provide evidence that the performance improvements shown in Table 3 are attributed to the proposed method, rather than being solely due to the use of more diverse data (8K labeled + 8K unlabeled).

[1] Simple baselines for image restoration, ECCV, 2022

[2] Toward blind flare removal using knowledge-driven flare-level estimator, TIP, 2024

---

> ### Author Rebuttal · Authors · 2026-03-30
>
> We sincerely thank Reviewer kx6P for the constructive feedback as "competitive results using significantly fewer labels". We address your concerns as below:
>
> ## **1. Fundamental Technical Innovation:**
>
> We agree that individual components are well-established in general low-level vision. However, naive Teacher-Student model application fails in our task, as degraded halo or streak shown in Figure 5. W/O both.
>
> Instead, our core innovation is a **problem-driven architecture** tailored to nighttime flares with a **close-up loop**:
>
> From Global Contrastive to Flare-Aware Disentanglement: Unlike standard contrastive learning with global representations, our strategy operates at the patch level and explicitly treats flare-contaminated regions as "hard negatives", forcing the network to decouple flare patterns from underlying textures.
>
> From Static Filtering to a Dynamic Quality Gate (NR-IQA Guided DR): We utilize NR-IQA not merely as a filter, but as a physical quality gate to break the error-accumulation loop. Dynamically refining pseudo-labels via momentum updates ensures the model learns from progressively reliable targets, relaxing the dependency on pseudo-label accuracy.
>
> Closed-Loop Synergy: contrastive loss and DR form a closed loop. The DR provides the high-quality, reliable targets  for the contrastive loss to function effectively, while the contrastive loss equips the student network with robust decoupling capability needed to generate better candidates for the DR. Table 2 validates this as removing either component drops PSNR from 27.620 to ~26.7.
>
> ## **2. Data Augmentation and Section B.1:**
>
> We thank the reviewer for this critical observation and clarify the confusion regarding the data augmentation strategy.
>
> First, we utilized the standard Flare7K++ synthesis pipeline (Sec. B.1) to ensure a strictly fair comparison with existing baselines (all same synthesis pipeline), which proves our performance gains entirely from the proposed semi-supervised framework, rather than an unfair data synthesis.
>
> Second, our "strong augmentations" aim to introduce strong perturbations to construct a challenging training task for better model generalization, which tackles with unseen degradation.  The strong augmentations combined with our Flare-aware Contrastive Loss, force the student network to learn more robust and invariant features rather than widening the domain gap.
>
> Finally, our approach achieves better real-world generalization compared to SOTAs. As shown in new added **Table A** below and Table 1 (generalization on FlareX) of manuscript, our approach gains well generalization performance, while visual comparisons (Fig. 8-9 in supplementary material) also validate this.
>
> **Table A: Cross-domain generalization performance on FlareReal600**
>
> | Method    | PSNR ↑     | SSIM ↑    | LPIPS ↓   |
> | :-------- | :--------- | :-------- | :-------- |
> | Flare7K   | 19.585     | 0.592     | 0.261     |
> | Flare7K++ | 19.890     | 0.599     | **0.257** |
> | SFHformer | 20.117     | 0.602     | 0.266     |
> | **Ours**  | **20.368** | **0.604** | 0.260     |
>
> ## **3. Evidence of Model-Agnostic Effectiveness:**
>
> Following your instruction, we conducted new experiments on baselines of U-Net, NAFNet, and Restormer in **Table B**, our framework improves all 5 evaluations on all baselines.
>
> **Table B: Quantitative results of different backbones with our semi-supervised framework (8K Labeled + 8K Unlabeled).**
>
> | Backbone    | Supervision Setting           | PSNR ↑     | SSIM ↑    | LPIPS ↓    | G-PSNR ↑   | S-PSNR ↑   |
> | :---------- | :---------------------------- | :--------- | :-------- | :--------- | :--------- | :--------- |
> | U-Net       | Limited-Sup. (8K Labeled)     | 25.098     | 0.856     | 0.0562     | 22.109     | 19.588     |
> | U-Net       | **Semi-Sup. (Ours: 8K + 8K)** | **25.531** | **0.874** | **0.0549** | **22.434** | **21.228** |
> | NAFNet      | Limited-Sup. (8K Labeled)     | 26.553     | 0.863     | 0.0529     | 23.437     | 21.321     |
> | NAFNet      | **Semi-Sup. (Ours: 8K + 8K)** | **27.112** | **0.892** | **0.0505** | **24.156** | **22.001** |
> | Restormer\* | Limited-Sup. (8K Labeled)     | 27.277     | 0.889     | **0.0497** | 23.970     | 21.557     |
> | Restormer\* | **Semi-Sup. (Ours: 8K + 8K)** | **27.431** | **0.898** | 0.0502     | **24.145** | **22.025** |
>
> _Regarding Flare-level net [2]:_ We have cited and introduced this paper in our revised manuscript. Since it is not open-sourced and for rebuttal fairness, we will connect with the authors later for testing in a future version.
>
> ## **4. Performance Improvements in Table 3:**
>
> Our method has better performance via problem-driven methodology, rather than 8K labeled + 8K unlabeled data setting. This is validated via SFSNiD\* with same data setting to obtain PSNR as 24.992, 2.628 lower than ours in Table 3 of manuscript. Meanwhile, our ablation studies in Table. 2 also validates that any module removal would cause PSNR drop between 0.909 and 1.11.

---

> > ### Author Rebuttal · Reviewer_kx6P · 2026-04-03
> >
> > The authors’ response addressed my concerns; however, the contribution of the paper is somewhat limited. Overall, I will raise my score to weak reject.

---

> > > ### Author Response · Authors · 2026-04-07
> > >
> > > We sincerely thank you for the time and effort reviewing our rebuttal. Your insightful suggestions were invaluable to further improve our manuscript's quality and clarity.

---

### Decision · Program_Chairs · 2026-04-30

**Decision:**

Accept (regular)

**Comment:**

This paper proposes Semi-LAR, a semi-supervised framework for nighttime flare removal that combines an adaptive pseudo-label repository with a flare-aware contrastive objective. The reviewers agreed that the paper addresses a practically important problem, especially given the difficulty of collecting paired real flare data, and that the empirical study is overall solid. The paper was also credited for competitive results across multiple benchmarks and for presenting a coherent framework for leveraging unlabeled data in this setting.

In the first round, the main concerns focused on the realism of the semi-supervised setting, the specific contribution of the NR-IQA-based pseudo-label filtering, the evidence supporting the model-agnostic claim, and the overall degree of technical novelty. After carefully reading the rebuttal and discussion, I believe the substantive technical concerns were addressed adequately. In particular, the additional experiments using real unlabeled flare images directly strengthened the semi-supervised claim, and the clarifications on the repository design, contrastive loss, and ablations resolved most reviewer questions. Three reviewers revised their assessments upward after rebuttal.

One reviewer nevertheless remained unconvinced about the level of novelty and continued to view the contribution as somewhat limited, despite acknowledging that the technical concerns had been addressed. I consider this a reasonable reservation, and the paper is indeed stronger in practical formulation and empirical support than in fundamental algorithmic novelty. On balance, however, I find the work acceptable. It addresses a relevant real-world problem and presents a coherent method with sufficient empirical support. I therefore recommend acceptance.